# Sirtuins as Key Regulators in Pancreatic Cancer: Insights into Signaling Mechanisms and Therapeutic Implications

**DOI:** 10.3390/cancers16234095

**Published:** 2024-12-06

**Authors:** Surbhi Chouhan, Anil Kumar, Naoshad Muhammad, Darksha Usmani, Tabish H. Khan

**Affiliations:** 1Lyda Hill Department of Bioinformatics, UT Southwestern Medical Center, Dallas, TX 75235, USA; 2Cecil H and Ida Green Center for Systems Biology, UT Southwestern Medical Center, Dallas, TX 75235, USA; 3Department of Systems Biology, Beckman Research Institute of City of Hope, Monrovia, CA 91016, USA; 4Department of Radiation Oncology, School of Medicine, Washington University, St. Louis, MO 63130, USA; 5Department of Ophthalmology, Washington University School of Medicine, St. Louis, MO 63130, USA; 6Department of Pathology and Immunology, Washington University School of Medicine, St. Louis, MO 63130, USA

**Keywords:** sirtuins, SIRT, deacetylase, pancreatic cancer, pancreatic ductal adenocarcinoma (PDAC), cancer signaling pathways, therapeutic interventions

## Abstract

Pancreatic ductal adenocarcinoma (PDAC) is a highly aggressive and often deadly cancer with limited treatment options. Sirtuins, a group of NAD^+^-dependent enzymes, have recently been identified as key players in PDAC progression and therapy resistance. Each sirtuin has a distinct role in PDAC biology. SIRT1 helps cancer cells survive chemotherapy, SIRT2 inhibits cell growth, and SIRT3 controls mitochondrial health, while SIRT4, SIRT5, SIRT6, and SIRT7 each influence metabolism, angiogenesis, and metastasis. This review explores how these sirtuins contribute to cancer development and highlights their potential as therapeutic targets. By specifically targeting sirtuin activity, new treatments could disrupt PDAC’s survival mechanisms, improve drug responses, and potentially offer more effective options for patients with this challenging cancer.

## 1. Introduction

Pancreatic cancer, particularly pancreatic ductal adenocarcinoma (PDAC), is one of the most lethal cancers due to its aggressive nature, late diagnosis, and poor responsiveness to conventional therapies [1,2,3]. Characterized by early metastasis, rapid tumor growth, and a highly immunosuppressive microenvironment, pancreatic cancer has a five-year survival rate of less than 10% [4,5]. Standard treatments such as chemotherapy and radiotherapy provide limited survival benefits due to the intrinsic resistance of PDAC cells to apoptosis and the tumor’s ability to rapidly acquire chemoresistance [6,7,8,9]. Conventional chemotherapeutic agents, including gemcitabine, nab-paclitaxel and FOLFIRINOX, remain the cornerstone of PDAC treatment, particularly for non-metastatic and advanced disease stages [10,11]. Among these, FOLFIRINOX demonstrates improved overall survival in metastatic PDAC patients compared to gemcitabine alone, yet the benefits are modest, and systemic toxicities remain a significant drawback [12]. Despite their use, less than 10% of patients respond robustly to gemcitabine-based therapies, as its short half-life and rapid clearance necessitate repeated dosing, which can induce severe side effects and therapeutic resistance [13]. This dire prognosis is largely due to complex genetic and epigenetic alterations in the signaling pathways that drive tumorigenesis, including KRAS mutations, TGF-β dysregulation, and aberrant Notch and Hedgehog signaling [14,15]. These signaling abnormalities not only promote cancer cell proliferation and invasion but also enable the tumor to evade immune surveillance, making it challenging to treat with current therapies. Thus, it is essential to explore the key signaling events that drive pancreatic cancer as critical regulators of its progression. In general, cancer signaling is highly complex, forming interconnected circuits that drive oncogenesis, tumor progression, and metastasis, often involving cross-talk between cascades like PI3K/AKT/mTOR, WNT/RAS/RAF/MEK/ERK, and TNFα/SMAD [16,17,18,19,20]. The dysregulation of these signaling pathways is frequently caused by genetic mutations, epigenetic alterations, metabolic alterations and aberrant post-translational modifications, which collectively contribute to the malignant phenotype [21,22,23,24,25,26,27]. Additionally, feedback loops and compensatory mechanisms often render cancer cells resilient to therapeutic interventions, further complicating the landscape of effective treatment strategies [28,29,30,31,32,33,34].

These genetic, signaling and microenvironmental factors also promote therapeutic resistance and hinder the efficacy of both conventional and targeted therapies in pancreatic cancer. For example, targeted agents like EGFR inhibitors (e.g., erlotinib), combined with gemcitabine, provide only modest survival benefits, extending one-year survival by approximately 23% [35,36]. To address these challenges, the development of novel biomarkers for early diagnosis and innovative molecular targets is critical. Recent research underscores the need for strategies that effectively deliver therapeutic agents to the tumor and its microenvironment [37,38,39]. In this context, nanoparticle-based drug delivery systems hold particular promise for overcoming these obstacles, enhancing drug accumulation within hypovascular tumor regions [40,41]. Nanomedicine-based approaches certainly improve drug delivery but face challenges in biodistribution [42,43,44]. Targeted therapies using small-molecule inhibitors and monoclonal antibodies aim to specifically disrupt oncogenic signaling pathways, such as those involving KRAS or proteins within the PI3K/mTOR and EGFR pathways, while sparing normal cells to minimize toxicity [45,46,47]. However, similar to nanomedicine-based therapeutic approaches, targeted therapy might exert pleiotropic effects. Despite extensive preclinical evaluations of key modulator-based targeted agents, their clinical translation remains challenging, emphasizing the urgent need for continued research into combination therapies and novel approaches that address the inherent heterogeneity and therapy resistance of PDAC. Additionally, a deeper understanding of signaling modulators and their upstream regulators is essential for advancing our knowledge of pancreatic cancer progression.

Accumulating evidence suggests that sirtuins are among the key regulators of multiple signaling pathways specific to pancreatic cancer, modulating essential cellular processes such as metabolic reprogramming, stress responses, and resistance to apoptosis, thereby significantly influencing tumor growth and survival. Sirtuins are a family of highly conserved proteins (SIRT1-SIRT7 in mammals) that function as NAD^+^-dependent deacetylases and ADP-ribosyltransferases [48,49,50]. These proteins regulate a wide range of cellular processes, including metabolism, aging and stress resistance, by modifying both histone and non-histone proteins [51]. Their roles are essential in the maintenance of cell survival and homeostasis through the modulation of epigenetic regulation, DNA repair, apoptosis and inflammation. In cancer, sirtuins play a dual role, acting as both tumor suppressors and tumor promoters, depending on the type of cancer and its cellular context [52,53,54,55]. Each sirtuin exerts distinct regulatory functions within these pathways, mediated by their varied subcellular localization and substrate specificity. For instance, SIRT1 influences the PI3K/AKT pathway by deacetylating key transcription factors, thereby modulating cell survival and chemoresistance [56,57,58,59,60]. SIRT2 plays a key role in cell cycle regulation by deacetylating proteins such as tubulin and histones, which ensure chromosomal stability [61]. SIRT2 has been implicated in KRAS-driven tumorigenesis, particularly through its role in maintaining genomic stability and regulating the RAS/RAF/MEK/ERK pathway [62]. In some cancers, such as gliomas, the downregulation of SIRT2 is associated with increased malignancy [63], suggesting its crucial role in cancer development. Mitochondrial sirtuins, including SIRT3, SIRT4 and SIRT5, are involved in cellular metabolism, redox balance and the oxidative stress response [64]. SIRT3, in particular, is often viewed as a tumor suppressor due to its ability to regulate mitochondrial function and reduce oxidative stress, which can inhibit cancer cell growth [65]. SIRT3 predominantly regulates oxidative phosphorylation and glycolysis, frequently in coordination with hypoxia-inducible factors [66,67]. SIRT4, in contrast, exerts tumor-suppressive effects by inhibiting aerobic glycolysis and attenuating the proliferative signals in the mTOR pathway [68,69,70]. SIRT5’s regulation of metabolic enzymes further adds another layer of complexity to cancer metabolism [71,72,73,74]. SIRT6 is another prominent tumor suppressor that regulates chromatin remodeling and transcription to enhance genome stability and counteract the metabolic reprogramming associated with cancer [75]; it often acts as a guardian of genomic integrity by influencing DNA repair pathways and modulating NF-κB signaling [76], which is crucial for inflammation-driven cancers [77]. The loss of SIRT6 is linked to increased glycolysis and tumor growth, particularly in pancreatic and colorectal cancers, underscoring its importance in cancer metabolism; meanwhile, SIRT7 generally functions as a tumor promoter, facilitating cancer cell survival and proliferation by regulating RNA polymerase I activity and rRNA synthesis [78,79]. Elevated SIRT7 expression is often associated with poor prognosis in cancers such as breast and liver cancer.

As the mechanistic exploration of sirtuins’ multifaceted roles in oncogenesis intensifies, growing scientific evidence has begun to unravel their critical involvement in the intricate pathophysiology of pancreatic cancer. Recent studies underscore the regulatory capacities of various sirtuin isoforms, revealing their significant influence on essential processes such as metabolic reprogramming, cellular stress response, and tumor progression within the pancreatic tumor microenvironment. This review provides a comprehensive exploration of the emerging roles of sirtuins in pancreatic cancer, focusing on how they interact with the key signaling pathways that influence critical processes such as tumor growth, drug resistance, and metabolic changes. By highlighting the complex and dual nature of sirtuins in both promoting and suppressing tumors, the review aims to uncover new therapeutic strategies that could disrupt cancer progression and enhance patient outcomes. It emphasizes the need for a deeper investigation of how sirtuins contribute to the development and survival of pancreatic cancer, advocating for targeted therapies that can specifically modulate sirtuin activity to overcome treatment resistance and improve the effectiveness of current therapies.

## 2. Sirtuins and Pancreatic Carcinogenesis

Pancreatic cancer, especially PDAC, is a highly aggressive malignancy with early metastasis, rapid tumor growth, and an immunosuppressive microenvironment, leading to poor prognosis and resistance to conventional therapies. Amid these challenges, emerging evidence suggests that the regulation of pancreatic cancer by sirtuins exemplifies a complex interplay between metabolic reprogramming, genomic integrity, and cellular stress responses. Sirtuins, such as SIRT1, SIRT2, and SIRT6, interact with crucial signaling pathways, influencing tumor metabolism, genomic stability, and the progression of ADM. SIRT1, for example, modulates hypoxia-inducible factors and exosomal circRNAs, contributing to both tumorigenesis and therapy resistance. In contrast, SIRT6 acts as a tumor suppressor by repressing glycolysis and maintaining chromatin stability, with its loss leading to increased metabolic dysregulation and genomic instability, driving tumor growth. The dual nature of sirtuins as both tumor promoters and suppressors highlights their complex, context-dependent roles in pancreatic carcinogenesis, making them critical targets for therapeutic intervention in PDAC. Based on their effects on pancreatic carcinogenesis, sirtuins can be tumor suppressors or tumor promoters, and each of these sirtuins will be examined in detail in the following sections.

### 2.1. SIRT1

SIRT1 has emerged as one of the most extensively studied sirtuins in pancreatic carcinogenesis. A landmark study offers a pioneering exploration of the complex role of SIRT1 in tumor differentiation and maintenance within the exocrine pancreas, shedding light on mechanisms that hold significant therapeutic promise. By examining the expression of SIRT1 and Dbc1 in both the normal exocrine pancreas and during ADM, a precursor stage to PDAC, the investigation aims to define the context-specific target genes regulated by SIRT1, thereby elucidating its complex role in pancreatic carcinogenesis [80]. This study reveals that in normal acinar cells, SIRT1 is co-expressed with KIAA1967 in the nucleus, yet during ADM, SIRT1 transiently shuttles from the nucleus to the cytoplasm. This nuclear-to-cytoplasmic transition of SIRT1 is crucial in promoting ADM, as experiments show that inhibiting SIRT1’s shuttling or activity during ADM suppresses this phenotypic change, highlighting the protein’s regulatory influence. The study also reveals that KIAA1967, a mediator of SIRT1 function, is downregulated in PDAC, affecting the sensitivity of PDAC cells to the SIRT1/2 inhibitor Tenovin-6. In contrast to its regulation of β-catenin acetylation in ADM, SIRT1’s effect on p53 acetylation in PDAC demonstrates its selective impact on different tumor regulators. Similarly, another study reveals that sirt1 regulates epithelial–mesenchymal transition (EMT) and tumor progression in pancreatic cancer, particularly through its interaction with Methyl-CpG binding domain protein 1 (MBD1) [81]. MBD1, a key player in transcriptional regulation and genomic stability, is significantly upregulated in pancreatic cancer tissues compared to adjacent normal tissues, which correlates with lymph node metastasis and poor patient survival. SIRT1 forms a functional complex with MBD1 and Twist on the CDH1 promoter, where it facilitates the suppression of E-cadherin transcription. This suppression is pivotal as it enhances the EMT process, allowing cancer cells to acquire invasive and migratory capabilities. Additionally, one study has revealed that SIRT1 directly suppresses β-catenin, a key oncogenic molecule upregulated by PAUF in pancreatic cancer cells, thereby inhibiting tumor cell proliferation [82]. The activation of SIRT1, either through its overexpression or by treatment with resveratrol, significantly reduces β-catenin protein levels and its transcriptional activity, leading to a decrease in β-catenin-mediated target genes such as cyclinD1. Importantly, SIRT1 facilitates the proteasomal degradation of β-catenin without requiring nuclear localization, and this process occurs independently of known β-catenin regulators like GSK-3β and Siah-1. The downregulation of β-catenin via SIRT1 activation effectively inhibits pancreatic cancer cell proliferation, highlighting the therapeutic potential of SIRT1 in targeting β-catenin-driven oncogenic pathways in pancreatic cancer.

An interesting study revealed that adiponectin deficiency significantly reduces pancreatic tumorigenesis in vivo, while adiponectin’s interaction with adipoR1 prevents apoptosis in both human and mouse pancreatic cancer cells, a mechanism central to which is the activation of the AMP-activated protein kinase (AMPK) and SIRT1 axis [83]. This study found that AMPK phosphorylation upregulates SIRT1, which in turn reciprocally phosphorylates AMPK, establishing a regulatory feedback loop that is crucial for pancreatic cancer cell survival. Furthermore, SIRT1 deacetylates PGC1α, promoting mitochondrial gene expression and reinforcing the anti-apoptotic effects. This signaling cascade, coupled with adiponectin’s elevation of β-catenin, underscores SIRT1’s critical role in linking metabolic and oncogenic pathways, positioning it as a key regulator in the survival and proliferation of pancreatic cancer cells.

A recent study highlighted the pivotal role of SIRT1 in modulating apoptotic pathways and enhancing cell survival, particularly in response to therapeutic interventions. Compound **19e**, a novel glucokinase activator, protects INS-1 pancreatic beta-cells from cytokine-induced apoptosis by increasing SIRT1 activity [84]. The compound **19e** facilitates the deacetylation of critical apoptotic regulators, thereby reducing cell death signals such as cleaved caspase-3 and poly (ADP-ribose) polymerase and promoting anti-apoptotic factors like B-cell lymphoma-2. This protective effect of **19e** is further corroborated by the attenuation of apoptotic signaling pathways, as evidenced by reduced cytochrome c release and the decreased expression of nuclear factor-κB p65 and inducible nitric oxide synthase. Additionally, SIRT1 also modulates the chemoresistance of pancreatic cancer through its interaction with hypoxic exosomal circular RNA (circRNA), particularly circZNF91 [85]. The circRNA, by binding competitively to miR-23b-3p, inhibits the microRNA’s regulatory effect on SIRT1, leading to increased levels of the deacetylase. The elevated SIRT1 then stabilizes the hypoxia-inducible factor 1 α (HIF1α) protein through deacetylation, which subsequently promotes glycolytic metabolism and contributes to the chemoresistance observed in pancreatic cancer cells. This process highlights the complex interplay between hypoxia, circRNA-mediated signaling and SIRT1 activity, as SIRT1 not only amplifies the effects of hypoxia but also sustains metabolic adaptations that facilitate resistance to gemcitabine. Moreover, the transcriptional upregulation of circZNF91 by HIF-1α further reinforces the feedback loop enhancing SIRT1 activity and glycolysis.

In summary, SIRT1 emerges as a pivotal regulator in pancreatic cancer through its multifaceted roles in tumor progression and resistance mechanisms. By modulating critical oncogenic processes such as epithelial–mesenchymal transition via its interactions with the MBD1-Twist-SIRT1 complex, SIRT1 not only influences tumor invasiveness but also impacts chemotherapy sensitivity. The capacity of SIRT1 to enhance cell viability and mitigate apoptotic signals, as demonstrated by compounds like **19e**, further emphasizes its role in protecting pancreatic beta-cells and cancer cells under stress conditions. These insights highlight SIRT1’s central position in pancreatic carcinogenesis, from its involvement in acinar-to-ductal metaplasia and acinar cell differentiation to its influence on chemoresistance. Consequently, targeting SIRT1 represents a promising therapeutic strategy for overcoming drug resistance and improving treatment outcomes in this aggressive malignancy, underscoring its potential as a strategic target for intervention in pancreatic cancer. Figure 1 summarizes an overview of the signaling pathways regulated by SIRT1 in pancreatic cancer.

### 2.2. SIRT2

Accumulating evidence suggests that SIRT2 might be a critical modulator in the carcinogenesis of pancreatic cancer, particularly through its aberrant expression in both benign and malignant tissues [86]. Immunohistochemical studies have demonstrated that the elevated cytoplasmic expression of SIRT2 in benign pancreatic tissue correlates with a reduced disease-free survival, suggesting that SIRT2’s role in pancreatic cancer may extend beyond tumor cells themselves by influencing the surrounding microenvironment. Although SIRT2’s expression pattern in malignant cells was less definitively linked to metastatic progression, its dysregulation likely contributes to the epigenetic landscape of tumorigenesis, interacting with other histone-modifying enzymes such as Jumonji domain 2 (JMJD2/KDM4) proteins. These findings indicate that SIRT2, alongside nuclear KDM4D expression, may serve as an important prognostic marker, particularly in the context of post-surgical recurrence. Consequently, SIRT2’s role in pancreatic carcinogenesis reflects its broader influence on cellular differentiation and epigenetic remodeling, highlighting its potential as a therapeutic target or risk indicator in clinical oncology. SIRT2 exerts significant influence on pancreatic cancer carcinogenesis by intricately regulating KRAS activity and its downstream oncogenic pathways. KRAS, a pivotal oncogene in pancreatic cancer, is frequently mutated, particularly at the G12D residue, driving the pathogenesis of PDAC in roughly 90% of cases. This mutation results in the constitutive activation of KRAS, leading to the aberrant stimulation of downstream signaling pathways, notably the RAF/MEK/ERK and PI3K/AKT cascades. During the carcinogenesis of pancreatic cancer, the imperative to exceed a critical threshold of KRAS activation for cellular transformation, combined with the fact that only a subset of KRAS mutant cells progress to malignancy, highlights the involvement of additional, albeit not fully elucidated, regulatory mechanisms. In this context, one study revealed that KrasG12D mice deficient in SIRT2 exhibit a markedly aggressive tumorigenic phenotype compared to their KRAS mutant, characterized by heightened cellular proliferation, increased KRAS acetylation and the augmented activation of downstream RAS signaling pathways [87]. Mechanistically, SIRT2 has been identified as a specific deacetylase for KRAS K147, with the acetylation status of this residue directly influencing KRAS’s active state. Similarly, another study underscores SIRT2’s critical role in modulating inflammatory responses and tissue recovery, highlighting its involvement in creating a microenvironment conducive to the development and progression of pancreatic cancer [88]. The enhanced tumorigenesis observed in SIRT2-/-KrasG12D mice, alongside increased KRAS acetylation and a markedly intensified systemic pro-inflammatory phenotype, became increasingly pronounced with advancing age.

Moreover, SIRT2 stabilizes Myc oncoproteins, which are critical drivers of pancreatic tumorigenesis. In pancreatic cancer cells, SIRT2 is upregulated by c-Myc, subsequently enhancing the stability of Myc proteins and promoting cancer cell proliferation [89]. Mechanistically, SIRT2 exerts its oncogenic influence by repressing the ubiquitin-protein ligase NEDD4, a key regulator of Myc protein degradation. By directly binding to the NEDD4 gene promoter and deacetylating histone H4 lysine 16, SIRT2 suppresses NEDD4 expression, thereby preventing the ubiquitination and proteasomal degradation of Myc. Furthermore, SIRT2 upregulates Aurora A expression, another factor that stabilizes Myc, amplifying its oncogenic effects. The inhibition of SIRT2 with small molecules reverses these effects by reactivating NEDD4 expression, promoting Myc degradation and suppressing pancreatic cancer proliferation. This underscores the potential therapeutic relevance of targeting SIRT2 in Myc-driven pancreatic malignancies.

Notably, there are limited reports addressing SIRT2-based inhibitors; however, the identification of NPD11033, a selective small-molecule inhibitor of SIRT2, highlights its remarkable specificity in targeting SIRT2 while sparing other sirtuins and zinc-dependent deacetylases [90]. NPD11033’s ability to induce a conformational change in the Zn-binding domain of SIRT2, creating a hydrophobic cavity behind the substrate-binding pocket, underscores its novel inhibitory mechanism. By stabilizing interactions with key residues, NPD11033 not only inhibits the deacetylase activity of SIRT2 but also impairs the proliferation of pancreatic cancer cells, as evidenced by the increased acetylation of its physiological substrate, eukaryotic translation initiation factor 5A (eIF5A). This finding positions NPD11033 as a promising lead in the development of SIRT2-targeted therapies in pancreatic carcinogenesis.

Further supporting the critical role of SIRT2 in pancreatic cancer, recent computational screenings have identified fluvastatin sodium as another potent inhibitor of SIRT2 [91]. Fluvastatin sodium, initially identified through computer simulations and molecular docking studies, demonstrated significant inhibitory activity against SIRT2, selectively binding to its active site with high docking scores and favorable interactions. Experimental validation through Western blotting and enzyme inhibition assays confirmed fluvastatin sodium’s potential to suppress pancreatic cancer metastasis by targeting SIRT2. Molecular dynamics simulations and binding free-energy calculations further elucidated the stability of the fluvastatin–SIRT2 complex, providing insights into the precise binding mechanisms. The suppression of SIRT2 activity by fluvastatin sodium offers a new avenue for therapeutic intervention, highlighting SIRT2 as a pivotal regulator of tumor progression and presenting a novel strategy for the targeted treatment of pancreatic cancer. Figure 2 provides a comprehensive summary of the signaling pathways modulated by SIRT2 in pancreatic cancer.

### 2.3. SIRT3

SIRT3, a pivotal mitochondrial deacetylase, exhibits a dual role in cancer biology, acting as either an oncogene or tumor suppressor depending on the tumor context. In pancreatic carcinoma, the expression of SIRT3 has been closely examined to elucidate its clinical significance. Immunohistochemical analyses across 79 pancreatic cancer patients revealed a pronounced downregulation of SIRT3 in cancer tissues relative to adjacent non-cancerous tissues, with statistical significance (*p* < 0.001) [92]. The decreased SIRT3 expression correlated strongly with an advanced T status and tumor stage (*p* < 0.001 and *p* = 0.013, respectively), suggesting that SIRT3 downregulation is intricately linked to more aggressive disease manifestations. Kaplan–Meier survival analysis further corroborated these findings, demonstrating a significant association between negative SIRT3 expression and poorer overall prognosis, thereby underscoring SIRT3’s potential as a critical biomarker in pancreatic cancer prognosis. In a comprehensive analysis of sirtuin family proteins, SIRT3 exhibited tumor-suppressive characteristics, particularly in PDAC [93]. The low cytoplasmic expression of SIRT3 was significantly associated with more aggressive tumor phenotypes and reduced patient survival, indicating its crucial role in mitigating tumor progression. The absence of SIRT3 also correlated with a shorter time to relapse and death in patients who did not receive chemotherapy, further underscoring its potential utility as a predictive biomarker for chemotherapeutic response.

Accumulating evidence offers profound insights into the mechanistic role of SIRT3 (SIRT3) in regulating the diverse signaling pathways implicated in pancreatic cancer. One such study reported that SIRT3 is pivotal in the carcinogenesis of pancreatic cancer by modulating the mitochondrial enzymatic functions critical for cellular metabolism and tumor progression [94]. Central to this process is the malate–aspartate shuttle, a vital system for transferring cytosolic NADH into mitochondria, thereby supporting glycolysis and rapid tumor cell proliferation. The study elucidates that mitochondrial glutamate oxaloacetate transaminase 2 (GOT2) undergoes acetylation at three specific lysine residues: K159, K185 and K404 by SIRT3. Acetylation at these sites enhances the interaction between GOT2 and malate dehydrogenase 2 (MDH2), facilitating the transfer of NADH into mitochondria and altering the mitochondrial NADH/NAD(+) redox balance. This modification not only promotes ATP production but also boosts NADPH generation, which mitigates oxidative stress by suppressing reactive oxygen species (ROS). Importantly, the acetylation of GOT2 at K159 is elevated in human pancreatic tumors and correlates with diminished SIRT3 expression. Similarly, another study claims that SIRT3 is associated with the regulation of pancreatic cancer progression through its modulation of the HIF-1α pathway [95]. Within the hypoxic tumor microenvironment typical of PDAC, HIF-1α promotes metabolic reprogramming, favoring glycolysis and angiogenesis, thereby supporting tumor growth and survival. SIRT3 acts as a tumor suppressor by destabilizing HIF-1α, thereby inhibiting these oncogenic processes. Profilin1 (Pfn1), a key actin-binding protein, has been shown to upregulate SIRT3 expression, further amplifying its inhibitory effect on HIF-1α. In PDAC cells with reduced Pfn1 expression, the downregulation of SIRT3 leads to increased HIF-1α stability, promoting a pro-tumorigenic hypoxic response. Conversely, the overexpression of Pfn1 restores SIRT3 levels, resulting in the destabilization of HIF-1α and subsequent suppression of tumor growth. Additionally, miR-421 directly represses SIRT3, leading to altered HIF-1α expression through the modulation of histone acetylation at H3K9, in pancreatic cancer [96]. This regulatory axis underscores the interplay between miR-421 and SIRT3 in manipulating the cellular hypoxic response and glycolytic activity, thereby promoting a more aggressive tumor phenotype.

SIRT3 has also been reported to regulate iron metabolism in pancreatic cancer, where dysregulated iron homeostasis is a hallmark of malignancy. SIRT3 modulates the activity of iron regulatory protein 1 (IRP1), a key regulator of cellular iron levels through its control of iron-responsive element (IRE)-containing genes [97]. In SIRT3-deficient cells, the heightened production of ROS enhances IRP1’s binding affinity to IREs, leading to the aberrant upregulation of iron-related genes such as the transferrin receptor (TfR1), which governs iron uptake and promotes cancer cell proliferation. The loss of SIRT3 disrupts iron homeostasis, resulting in iron overload and dysregulated TfR1 expression, phenomena observed both in SIRT3-null cells and in the pancreatic tissues of SIRT3-deficient mice. Importantly, SIRT3 acts as a tumor suppressor by attenuating TfR1 expression and iron uptake, thereby curbing the proliferative capacity of pancreatic cancer cells. This underscores a novel mechanistic link between SIRT3’s tumor-suppressive function and its regulation of cellular iron metabolism, positioning SIRT3 as a critical metabolic modulator in pancreatic carcinogenesis. A recent report suggested that SIRT3 is also involved in the regulation of the electron transport chain (ETC) in pancreatic cancer, particularly through its modulation of mitochondrial complex II (CII) activity [98]. As a key component of mitochondrial bioenergetics, SIRT3 maintains the functionality of CII, also known as succinate dehydrogenase (SDH), which is integral to the transfer of electrons to ubiquinone in the ETC. In pancreatic cancer cells, the dysregulation of SIRT3 correlates with impaired CII activity, contributing to mitochondrial dysfunction and promoting tumor progression. SIRT3 loss enhances the expression of HIF-1α, a driver of hypoxia-mediated metabolic reprogramming in cancer, leading to an increased reliance on glycolysis and reduced oxidative phosphorylation. The diminished activity of SDH subunits C and D, along with SIRT3 downregulation, disrupts electron flow within the ETC, resulting in elevated ROS production and apoptosis resistance. However, pharmacological agents like chrysin nanoparticles (CCNPs), which restore CII activity and SIRT3 expression, have demonstrated significant potential in reactivating ETC function, thereby inducing apoptosis and reducing the viability of pancreatic cancer cells. By preserving mitochondrial integrity and curbing aberrant ETC regulation, SIRT3 serves as a critical suppressor of oncogenic metabolism, positioning it as a key therapeutic target in pancreatic ductal adenocarcinoma.

Moreover, SIRT3 assumes a critical role in modulating glycolysis in pancreatic cancer, particularly by counteracting the tumor-promoting consequences of dysregulated glycolytic activity. In pancreatic cancer, the transcriptional silencing of SIRT3 by zinc finger E-box binding homeobox-1 (ZEB1) in cooperation with the methyl-CpG-binding domain protein 1 (MBD1) promotes a metabolic shift towards glycolysis. ZEB1, through its interaction with MBD1, suppresses SIRT3 expression, thereby enhancing glycolytic flux and reducing mitochondrial respiration [99]. The suppression of SIRT3 is central to the metabolic reprogramming that fuels the rapid proliferation of cancer cells via enhanced glucose uptake and lactate production. By modulating SIRT3 expression, ZEB1 not only drives aerobic glycolysis but also contributes to the broader metabolic flexibility and survival of pancreatic cancer cells under the hypoxic and nutrient-poor conditions leading to alterations in the tumor microenvironment.

Interestingly, a recent study revealed that the zinc finger protein ZMAT1, downregulated in PDAC, exerts its tumor-suppressive effects via the SIRT3/p53 signaling axis [100]. The overexpression of ZMAT1 inhibits pancreatic cancer cell proliferation by upregulating SIRT3, which subsequently activates p53, leading to cell cycle arrest and reduced tumorigenesis. The activation of SIRT3 by ZMAT1 underscores its regulatory role in maintaining mitochondrial integrity and metabolic homeostasis, essential in curbing the uncontrolled growth of PDAC cells. Thus, targeting SIRT3 in pancreatic cancer represents a critical axis in regulating tumorigenesis, given its involvement in mitochondrial homeostasis, oxidative stress regulation and metabolic adaptation. In PDAC, the dysregulation of SIRT3 has been linked to enhanced tumor proliferation, invasion and metastasis, through miRNA-708-5p [101]. This miRNA is upregulated in PDAC and is associated with poor patient prognosis. Functionally, miRNA-708-5p directly targets and suppresses SIRT3 expression, thereby disrupting its tumor-suppressive role. The depletion of SIRT3 facilitates a metabolic shift towards increased glycolysis, elevated mitochondrial ROS production and diminished autophagy, all of which favor tumor progression. Thus, re-establishing SIRT3 function in PDAC cells could reverse these oncogenic effects, making SIRT3 a promising therapeutic target. Additionally, therapeutic strategies such as Tris (dibenzylideneacetone) dipalladium (Tris DBA), a small-molecule palladium complex, may offer significant promise in targeting SIRT3-mediated pathways. Tris DBA has demonstrated potent antiproliferative effects across various malignancies, including pancreatic cancer. Its ability to modulate autophagy and mitigate mitochondrial ROS generation, particularly via the SIRT1- and SIRT3-mediated pathways, highlights its potential utility in disrupting tumor growth [102]. By inhibiting key signaling cascades like JNK, ERK and p38 MAPK and blunting NLRP3 inflammasome activation, Tris DBA acts on multiple levels of tumor pathophysiology. In PDAC, where SIRT3 downregulation exacerbates metabolic dysregulation and oxidative damage, the combined approach of SIRT3 reactivation and inhibition of oncogenic signaling presents a novel, multifaceted strategy for suppressing cancer progression and improving patient outcomes.

In conclusion, targeting SIRT3 emerges as a promising therapeutic approach in pancreatic cancer, given its multifaceted role in regulating mitochondrial metabolism, oxidative stress and key oncogenic pathways. The repression of SIRT3 contributes to the metabolic reprogramming of pancreatic cancer cells, enhancing glycolytic flux and supporting tumor progression. By restoring SIRT3 expression, it is possible to modulate mitochondrial function, induce apoptosis and disrupt hypoxia-driven oncogenic survival mechanisms, positioning SIRT3 as both a potent tumor suppressor and a critical biomarker for pancreatic cancer prognosis. The complex interplay of SIRT3 within the ZMAT1–SIRT3–p53 axis, along with its involvement in pathways such as the malate–aspartate shuttle and hypoxia response, highlights its therapeutic potential. Figure 3 offers a detailed overview of the signaling pathways regulated by SIRT3 in pancreatic cancer.

### 2.4. SIRT4 in Pancreatic Cancer

SIRT4 (SIRT4) plays a significant role in cancer biology, primarily through its regulatory functions in metabolic processes and cellular stress responses. As a mitochondrial protein, SIRT4 modulates key metabolic pathways, particularly those involving amino acids like glutamine [103]. By inhibiting glutamine metabolism, SIRT4 restricts the availability of substrates that cancer cells rely on for rapid proliferation, thus acting as a metabolic checkpoint. Additionally, SIRT4 is linked to the regulation of autophagy, a cellular process that can promote survival in low-nutrient conditions while facilitating the elimination of damaged organelles and proteins [104]. In some cancers, SIRT4-induced autophagy can suppress tumor growth, while in others, it may contribute to survival under adverse conditions. Furthermore, by influencing [105], SIRT4 modulates the cellular response to oxidative stress, helping to maintain redox balance. Some studies suggest that SIRT4 functions as a tumor suppressor by activating the pathways associated with cell cycle regulation and apoptosis, such as the phosphorylation of p53, positioning it as a potential target for therapeutic intervention [106,107]. Moreover, SIRT4’s expression levels have been correlated with clinical outcomes in various cancers, where lower levels often indicate more aggressive disease and poorer prognosis. Although reports exploring the role of SIRT4 (SIRT4) in pancreatic cancer are limited, emerging evidence suggests that this mitochondrial protein plays a pivotal role in regulating metabolic processes in pancreatic β-cells and PDAC. In the context of β-cell function, global knockout studies in mice have shown that SIRT4 deficiency leads to increased glucose- and leucine-stimulated insulin secretion, culminating in age-related glucose intolerance and insulin resistance [108]. However, intriguingly, the β-cell-specific ablation of SIRT4 did not replicate these alterations, suggesting that SIRT4’s regulatory influence on nutrient-stimulated insulin secretion may extend beyond the β-cells themselves. This indicates a more systemic role for SIRT4 in orchestrating insulin homeostasis, perhaps through its interaction with other metabolic tissues, underscoring the complexity of its function in glucose metabolism.

In pancreatic cancer, SIRT4 assumes a tumor-suppressive role by modulating autophagy, a key process involved in cancer cell survival and growth. A recent report has revealed that SIRT4 promotes autophagy in PDAC by inhibiting glutamine metabolism, subsequently triggering the phosphorylation of p53, a tumor suppressor protein, via the AMPKα pathway [109]. This SIRT4/AMPKα/p53 axis is crucial for limiting pancreatic tumorigenesis, with SIRT4 acting as a gatekeeper that curbs tumor growth by fostering autophagic processes. Another report suggests that SIRT4 is a downstream target of UHRF1 (ubiquitin-like with plant homeodomain and ring finger domains 1), an epigenetic modifier known to mediate the silencing of tumor suppressor genes [110]. In pancreatic cancer cells, UHRF1 overexpression promotes aerobic glycolysis and supports tumor growth and metastasis, in part by stabilizing HIF1α and enhancing the transcription of glycolytic genes. However, SIRT4 negatively regulates these oncogenic processes, inhibiting glycolysis, cell proliferation and tumor expansion. Moreover, the SIRT4–UHRF1 axis plays a critical role in the regulation of PDAC, with recent studies revealing that the overexpression of the scaffold protein WDR79 in PDAC promotes tumor growth and motility by primarily enhancing aerobic glycolysis [111]. However, the knockdown of WDR79 significantly inhibits these oncogenic processes by upregulating SIRT4 through the suppression of UHRF1 expression. UHRF1, an epigenetic regulator known for silencing tumor suppressor genes, negatively impacts SIRT4 levels, thereby facilitating cancer cell proliferation and metabolic reprogramming. The depletion of WDR79 disrupts this axis, restoring SIRT4’s inhibitory influence on glycolysis and halting PDAC progression.

Additionally, recent findings have underscored SIRT4’s critical involvement in maintaining mitochondrial homeostasis. SIRT4 deacetylates lysine 547 of SEL1L, leading to an upregulation of the E3 ubiquitin ligase HRD1, a key component of the endoplasmic reticulum-associated protein degradation (ERAD) pathway [112]. The HRD1–SEL1L complex destabilizes ALKBH1, a mitochondrial protein, impairing the transcription of mitochondrial DNA-encoded genes and inducing mitochondrial dysfunction. This disruption of mitochondrial integrity curbs cancer cell proliferation. Additionally, SIRT4 emerges as a critical tumor suppressor in PDAC, with its regulatory role extending across mitochondrial metabolism, cellular proliferation and cancer progression. The novel WDR79–UHRF1–SIRT4 axis underscores the centrality of SIRT4 in inhibiting the aerobic glycolysis that fuels PDAC malignancy, positioning it as a pivotal antagonist of cancer metabolism. The inverse correlation between UHRF1 and SIRT4 expression further amplifies the therapeutic potential of targeting this axis to disrupt PDAC progression. Importantly, the identification of Entinostat, a potent SIRT4 stimulator, highlights a promising avenue for clinical intervention, as its ability to upregulate SIRT4 expression significantly impairs PDAC growth both in vitro and in vivo. This multifaceted regulation of cancer metabolism and mitochondrial integrity establishes SIRT4 as not only a metabolic gatekeeper but also a potent therapeutic target, warranting further exploration to refine treatment strategies for this aggressive malignancy. Figure 4 presents a comprehensive depiction of the signaling pathways modulated by SIRT4 in pancreatic cancer.

### 2.5. SIRT5 in Pancreatic Cancer

SIRT5 plays a multifaceted role in cellular metabolism and is emerging as a key regulator in various cancers, including pancreatic cancer. SIRT5 is primarily involved in regulating metabolic processes such as glutamine metabolism, fatty acid oxidation and ROS detoxification through lysine desuccinylation, demalonylation and deglutarylation [113,114]. In pancreatic cancer, dysregulated metabolic pathways are critical for tumor growth and survival and SIRT5 may contribute to the adaptation of cancer cells to the nutrient-deprived tumor microenvironment [115]. Clinical studies have demonstrated that high SIRT5 expression is associated with reduced tumor cell proliferation and improved patient prognosis in PDAC [116]. In mouse models, the genetic ablation of Sirt5 promotes acinar-to-ductal metaplasia, precursor lesion formation and heightened tumor growth, correlating with poor survival outcomes. Mechanistically, SIRT5 loss enhances glutamine and glutathione metabolism by activating GOT1 through acetylation, a pathway critical for sustaining PDAC’s metabolic demands. The discovery of MC3138, a selective SIRT5 activator, provides a promising therapeutic strategy by mimicking the effects of SIRT5 overexpression, reducing nucleotide pools and sensitizing PDAC cells and organoids to gemcitabine. These findings position SIRT5 as a crucial metabolic gatekeeper and its activation represents a novel avenue for targeted therapy in pancreatic cancer. Thus, SIRT5 has gained recognition as a critical tumor suppressor in PDAC, with its depletion markedly accelerating tumorigenesis and metabolic reprogramming. Despite the promising evidence of SIRT5’s regulatory influence on key metabolic pathways such as glutamine and glutathione metabolism, there remains a significant gap in comprehensive studies exploring its precise role in PDAC progression. The paucity of research highlights the urgent need for further investigations of SIRT5’s function, as its modulation could present a novel therapeutic strategy by disrupting the metabolic dependencies that sustain pancreatic cancer cells. Given the metabolic plasticity of PDAC, SIRT5’s role in regulating noncanonical metabolic pathways positions it as a promising therapeutic target, with its activation offering a potential strategy to suppress tumor growth and improve the efficacy of existing treatments. Figure 5 provides an in-depth illustration of the signaling pathways regulated by SIRT5 in pancreatic cancer.

### 2.6. SIRT6 in Pancreatic Cancer

SIRT6 is increasingly recognized for its pivotal role in cancer biology, particularly given its multifaceted involvement in regulating cellular metabolism, maintaining genomic stability and modulating inflammatory responses [77]. Ubiquitously expressed across various tissues, SIRT6 orchestrates chromatin dynamics through deacetylation, promoting transcriptional repression and facilitating the cellular response to DNA damage [117]. Its unique enzymatic profile, characterized by significantly lower deacetylation activity compared to other Sirtuins, highlights a complex biochemical landscape in which SIRT6 functions primarily through its deacylation capabilities [118]. Notably, SIRT6 enhances DNA repair mechanisms, particularly in response to double-strand breaks, by recruiting critical repair factors such as 53BP1 and BRCA1, thereby underscoring its essential contribution to maintaining genomic integrity. Moreover, SIRT6’s ability to modulate inflammatory responses adds another layer of complexity to its role in cancer. It can act both as an activator and repressor of inflammation, influencing the expression of pro-inflammatory cytokines while concurrently suppressing pathways like NF-κB that drive inflammatory responses [119]. This dual functionality positions SIRT6 as a critical determinant of cell fate, particularly in the context of tumorigenesis. Evidence suggests that SIRT6 can modulate the expression of pro- and anti-apoptotic factors, thereby influencing cancer cell survival in a context-dependent manner [120,121]. For instance, in the presence of DNA damage, SIRT6 may induce apoptosis via the mono-ADP-ribosylation of p53 and p73, highlighting its potential use as both a tumor promoter and suppressor depending on the cellular context [122,123].

Accumulating evidence suggests that the dual functions of SIRT6 as a regulator of metabolic processes and a guardian of genomic stability have positioned it as a key regulator in pancreatic cancer. One significant pathway involves the interaction between SIRT6 and Krüppel-like factor 10 (KLF10), a known tumor suppressor. In KLF10-depleted PDAC models, SIRT6 plays a pivotal role in reversing the malignant phenotypes, including increased EMT and glycolysis, which are characteristic of aggressive tumor progression. SIRT6 overexpression was found to mitigate the migratory and glycolytic tendencies of KLF10-depleted cells, highlighting its regulatory influence on both glucose homeostasis and EMT [124]. The ability of SIRT6 to modulate these critical oncogenic pathways, particularly via the NF-κB and HIF1α axes, positions it as an essential mediator in preventing distant metastasis in PDAC, as demonstrated by improved the survival rates in murine models upon SIRT6 activation. This novel KLF10/SIRT6 signaling pathway presents promising therapeutic potential by ameliorating tumor metastasis through the coordinated regulation of metabolic and migratory processes. In addition to its metabolic regulation, SIRT6 exerts a profound influence on the epigenetic landscape of PDAC. Recent studies have revealed that SIRT6 inactivation accelerates PDAC progression through the upregulation of Lin28b, a key modulator of the let-7 microRNA pathway. The loss of SIRT6 leads to histone hyperacetylation at the Lin28b promoter, facilitating Myc recruitment and the subsequent induction of oncogenes such as HMGA2, IGF2BP1 and IGF2BP3 [125]. This dysregulation of the epigenetic program not only promotes tumor growth but also defines a distinct molecular subset of PDAC with reduced SIRT6 expression, linked to poor prognosis.

Interestingly, recent studies have revealed that the basal subtype of PDAC, which is associated with poorer survival outcomes, exhibits a unique sensitivity to transcriptional inhibition through the targeting of cyclin-dependent kinases 7 and 9 (CDK7 and CDK9). Notably, this sensitivity correlates with the inactivation of the integrated stress response (ISR), which results in an increased rate of global mRNA translation. SIRT6 has been identified as a critical modulator of a constitutively active ISR by binding to activating transcription factor 4 (ATF4) within nuclear speckles, thereby enhancing its stability and preventing proteasomal degradation [126]. The loss of SIRT6 not only delineates the basal PDAC subtype but also disrupts ATF4 stability and renders the ISR nonfunctional, thereby exacerbating the vulnerability of these tumors to CDK7 and CDK9 inhibitors. This mechanistic insight underscores the potential use of SIRT6 as a therapeutic target in the treatment of aggressive PDAC forms, paving the way for novel intervention strategies that exploit transcriptional vulnerabilities. Moreover, the expression levels of SIRT6 in pancreatic cancer tissues and cell lines have been shown to correlate inversely with patient prognosis, highlighting its role as a tumor suppressor. In the context of PDAC, SIRT6 elevates the expression of pro-inflammatory cytokines and chemokines, such as IL-8 and TNF, thereby fostering an inflammatory phenotype in pancreatic cancer cells. This enhancement is mediated through SIRT6’s enzymatic activity, which increases intracellular levels of ADP-ribose, subsequently activating the Ca^2+^ channel TRPM2 [127]. The interplay between SIRT6 and TRPM2 leads to heightened Ca^2+^ responses that are crucial for the expression of TNF and IL-8. Moreover, SIRT6 facilitates the nuclear accumulation of the Ca^2+^-dependent transcription factor nuclear factor of activated T cells (NFAT), linking calcium signaling to pro-inflammatory gene expression. The inhibition of calcineurin, which diminishes NFAT activity, effectively reduces the expression of these inflammatory cytokines in SIRT6-overexpressing cells. Consequently, SIRT6’s multifaceted role in promoting the synthesis of Ca^2+^ mobilizing second messengers and regulating Ca^2+^-dependent transcription factors underscores its potential use as a therapeutic target to mitigate cancer-induced inflammation, angiogenesis and metastasis in pancreatic cancer.

SIRT6 has increasingly been recognized as a promising therapeutic target in pancreatic cancer, particularly due to its multifaceted role in modulating tumorigenesis, angiogenesis and responses to chemotherapy. Recent discoveries related to novel SIRT6 inhibitors, including salicylate-based compounds and pyrrole-pyridinimidazole derivatives, have demonstrated their capacity to enhance histone acetylation while inhibiting pro-inflammatory cytokine production [128,129]. These inhibitors not only sensitize pancreatic cancer cells to gemcitabine, a standard chemotherapeutic agent, but also demonstrate potent anti-angiogenic effects by suppressing key angiogenesis and hypoxia-related proteins, such as VEGF and HIF-1α, thereby inhibiting endothelial cell migration and tube formation. In xenograft models, these compounds have shown promise in downregulating CD31, a marker of angiogenesis, suggesting their potential utility in controlling tumor vascularization and progression. Notably, the novel allosteric inhibitor compound **11e** has demonstrated remarkable inhibitory potency, with an IC50 value of 0.98 ± 0.13 μmol/L, showcasing its efficacy in selectively targeting SIRT6 while sparing other histone deacetylases [130]. This selectivity is critical given the structural similarities that challenge the development of conventional inhibitors. Moreover, structural analyses of its mechanism, including interactions within the allosteric site, highlight the sophisticated design of this compound. Importantly, the antimetastatic capabilities of compound **11e** were confirmed in both pancreatic cancer cell lines and mouse models, marking a significant milestone in the exploration of SIRT6 inhibitors for therapeutic application. In parallel, the discovery of JYQ-42, another potent allosteric modulator, further exemplifies the innovative strategies being employed to target SIRT6 [131]. By leveraging a reversed allosteric strategy to uncover the cryptic site Pocket Z, JYQ-42 not only achieves the selective inhibition of SIRT6 but also significantly curtails cancer cell migration and pro-inflammatory cytokine production, both of which are pivotal in the metastatic cascade. The distinct mechanism of action conferred by allosteric modulation paves the way for the development of therapeutics that can precisely navigate the complexities of SIRT6’s role in cancer progression. In conclusion, the capacity of SIRT6 to function as a tumor suppressor, particularly through its modulation of pathways such as Lin28b, highlights its therapeutic promise for specific PDAC subtypes that rely on these epigenetic mechanisms for growth and survival. As the complexities of SIRT6’s dual role continue to unravel, the development of small-molecule inhibitors and activators stands to enhance our understanding of its biological functions. This strategic modulation of SIRT6 activity may pave the way for innovative cancer therapies, enabling personalized treatment regimens that exploit the unique regulatory mechanisms SIRT6 offers in combating this formidable malignancy. Ultimately, advancing our comprehension of SIRT6 could catalyze significant progress in the fight against pancreatic cancer, with the potential to improve outcomes for patients afflicted by this lethal disease. Figure 6 offers a detailed representation of the signaling pathways governed by SIRT6 in pancreatic cancer.

### 2.7. SIRT7 in Pancreatic Cancer

SIRT7 has emerged as a multifaceted regulator of tumorigenesis, exhibiting duality in its roles as both a pro-tumorigenic and tumor-suppressive entity. Unlike other SIRT members, such as SIRT1 and SIRT6, which also demonstrate similar dichotomies, SIRT7 is distinguished by its pivotal involvement in nucleolar functions and a pronounced association with RNA metabolism and processing, an attribute that is notably unique among the sirtuin family [79]. This complex multi-enzymatic nature of SIRT7 endows it with the ability to interact with a diverse array of molecular targets, thereby influencing multiple pathways involved in cancer progression. Notably, SIRT7 expression is upregulated in pancreatic cancer cells, correlating with adverse prognostic outcomes and a heightened resistance to gemcitabine, a cornerstone of pancreatic cancer treatment. A comprehensive analyses, integrating RNA-sequencing and ATAC-sequencing data, identified GLUT3 as a potential downstream target of SIRT7 [127]. Mechanistically, SIRT7 facilitates the desuccinylation of histone H3 at lysine 122 within the enhancer region of GLUT3, thereby modulating its transcriptional activity. Subsequent investigations revealed that GLUT3 not only contributes to the transport of gemcitabine in breast cancer cells but also mediates gemcitabine sensitivity in PC; the knockdown of GLUT3 diminished the chemotherapeutic efficacy of gemcitabine, while SIRT7 knockdown enhanced sensitivity, an effect that was abrogated upon GLUT3 knockdown. Moreover, fasting mimicking protocols induced SIRT7 expression in pancreatic cancer cells, yet counterintuitively, SIRT7 deficiency augmented gemcitabine sensitivity by upregulating GLUT3 levels. Utilizing a mouse xenograft model, we further corroborated the influence of SIRT7 deficiency on gemcitabine sensitivity under fasting conditions. Another investigation utilizing mass spectrometry unveiled SIRT7’s interaction with O-GlcNAc transferase (OGT), which plays a pivotal role in the post-translational modification of SIRT7 through O-GlcNAcylation. This modification not only stabilizes SIRT7 by inhibiting its association with REGγ, thus preventing proteasomal degradation, but also facilitates the hypoacetylation of histone H3 at lysine 18 (H3K18) via SIRT7, culminating in the transcriptional repression of several tumor suppressor genes [132]. Moreover, the O-GlcNAcylation of SIRT7 at serine 136 (S136) is indispensable for maintaining both its protein stability and deacetylation capacity. In both in vivo and in vitro settings, the attenuation of SIRT7 O-GlcNAcylation at S136 was observed to impede tumor progression, thereby underscoring the significance of this modification in the context of PDAC.

Interestingly, recent studies elucidating the expression profiles of sirtuins in PDAC revealed that low levels of nuclear SIRT7 correlate with more aggressive tumor phenotypes and poorer patient outcomes, thus establishing it as a significant biomarker for disease prognosis. Further investigations demonstrated that SIRT7 inhibits EMT in pancreatic cancer cells by transcriptionally repressing key target genes such as COL4A1 and SLUG, which are implicated in enhancing metastatic capabilities [133]. This study, utilizing siRNA and overexpression techniques, showed that SIRT7 overexpression led to decreased cell proliferation, migration and invasion, while its knockdown had the opposite effect. Moreover, the regulatory influence of SIRT7 extends to various cancer-related signaling pathways, highlighting its multifaceted role in modulating the tumor microenvironment [93]. The ability of SIRT7 to stabilize and inhibit the expression of pro-tumorigenic factors underscores its potential as a therapeutic target. Consequently, the development of SIRT7-based inhibitors may provide a novel strategy for enhancing treatment efficacy in pancreatic cancer, a malignancy characterized by its notorious resistance to conventional therapies and poor prognosis. In summary, the exploration of SIRT7 as a therapeutic target in pancreatic cancer elucidates its multifaceted role in tumorigenesis and positions it as a pivotal player in developing innovative treatment strategies. The regulatory influence of O-GlcNAcylation on SIRT7 underscores its significance in modulating the critical pathways associated with tumor aggressiveness, thereby revealing opportunities for targeted interventions. Furthermore, the promising potential of SIRT7-based inhibitors, particularly when combined with fasting protocols, may synergistically enhance the efficacy of established chemotherapeutics like gemcitabine. Collectively, these insights advocate for a more nuanced understanding of SIRT7’s mechanistic pathways, facilitating the identification of effective therapeutic strategies that could significantly improve clinical outcomes for patients grappling with this formidable malignancy. Figure 7 provides an intricate depiction of the signaling pathways regulated by SIRT7 in pancreatic cancer.

## 3. SIRT-Targeted Drug Responses in Pancreatic Cancer

Sirtuins have emerged as promising prognostic biomarkers in pancreatic cancer due to their diverse roles in tumorigenesis and correlation with patient outcomes. SIRT1 is highly expressed in poorly differentiated carcinomas, making it a strong candidate for prognostic evaluation. Similarly, SIRT3 and SIRT6 exhibit tumor-suppressive functions, with their low expression associated with tumor progression and poor survival, establishing their potential as reliable biomarkers. SIRT7, whose expression correlates with aggressive tumor phenotypes and poor outcomes, also holds significant promise in predicting disease prognosis. While the roles of SIRT2, SIRT4, and SIRT5 in pancreatic cancer progression are evident, their utility as biomarkers is not yet fully established. Table 1 outlines the specific roles of sirtuins in pancreatic cancer tumorigenesis and highlights their potential as prognostic biomarkers for the disease. Beyond their potential as biomarkers, sirtuins also offer a unique avenue for therapeutic intervention. Targeting sirtuins through pharmacological modulation has shown promise in altering key oncogenic pathways in pancreatic cancer. For instance, strategies aimed at restoring the tumor-suppressive functions of SIRT3 or enhancing SIRT4 activity could reverse metabolic reprogramming and tumor survival mechanisms. Additionally, inhibiting SIRT1 and SIRT7 activity, which contribute to chemoresistance and aggressive phenotypes, respectively, could sensitize tumors to conventional treatments like chemotherapy. The dual utility of sirtuins as both biomarkers and therapeutic targets underscores their pivotal role in advancing personalized medicine approaches for pancreatic cancer, offering hope for improved prognosis and treatment outcomes in this devastating disease.

Targeting the sirtuin family offers a promising approach to developing innovative therapeutics to combat pancreatic cancer, a malignancy known for its resistance to standard treatments and poor prognosis. SIRT1, as a key modulator of apoptotic pathways and cell survival, plays a pivotal role in pancreatic cancer’s resistance to chemotherapy. By enhancing the activity of pro-survival factors and inhibiting apoptotic signals, SIRT1 supports tumor progression. Furthermore, its interaction with hypoxic exosomal circular RNA (circZNF91) and its role in sustaining metabolic adaptations under hypoxic conditions present a compelling case for the development of SIRT1 inhibitors. The therapeutic targeting of SIRT1 may disrupt this feedback loop, attenuating chemoresistance and improving the efficacy of drugs like gemcitabine. SIRT2, though less studied in pancreatic cancer, has emerged as a potential therapeutic target with the discovery of NPD11033, a selective small-molecule inhibitor that induces a conformational change in SIRT2, inhibiting its activity. Given their ability to impair tumor cell proliferation by modulating the acetylation of critical substrates like eIF5A, SIRT2 inhibitors represent a novel avenue for pancreatic cancer therapy. This approach may be particularly beneficial in combination with other treatments aimed at targeting the metabolic and proliferative pathways central to pancreatic tumorigenesis. The tumor-suppressive functions of SIRT3 in pancreatic cancer are also gaining attention, particularly regarding its regulation of cellular metabolism. SIRT3 inhibition by miRNA-708-5p in PDAC leads to enhanced glycolysis, increased oxidative stress and diminished autophagy, all of which promote tumor growth. Re-establishing SIRT3 activity through therapeutic interventions such as Tris DBA, a palladium-based compound, offers a promising strategy for disrupting these oncogenic pathways. Tris DBA’s capacity to modulate autophagy and reduce mitochondrial ROS highlights the potential ability of targeting SIRT3 to reverse the metabolic dysregulation commonly observed in pancreatic cancer.

SIRT4’s role in inhibiting aerobic glycolysis, a hallmark of cancer metabolism, further supports its therapeutic potential in pancreatic cancer. The identification of the WDR79–UHRF1–SIRT4 axis highlights the centrality of SIRT4 in antagonizing PDAC malignancy. Entinostat, a potent stimulator of SIRT4, demonstrates the promise of using SIRT4-targeted therapies to impair tumor growth, offering a novel clinical intervention for this aggressive disease. SIRT5, another critical regulator of metabolic processes in pancreatic cancer, modulates glutamine and glutathione metabolism, crucial for sustaining the tumor’s metabolic demands. The activation of SIRT5 through selective agents like MC3138 has shown efficacy in reducing nucleotide pools and sensitizing PDAC cells to gemcitabine, underscoring the potential use of SIRT5 activators as an effective therapeutic strategy in combination with existing treatments. SIRT6, recognized for its multifaceted role in tumorigenesis, angiogenesis and chemoresistance, has garnered increasing interest in pancreatic cancer research. Novel inhibitors, such as salicylate-based compounds and pyrrole-pyridinimidazole derivatives, enhance histone acetylation and suppress pro-inflammatory cytokine production, thereby sensitizing cancer cells to chemotherapy while inhibiting angiogenesis. The discovery of potent SIRT6 inhibitors like compound **11e**, with its remarkable selectivity and anti-metastatic properties, highlights the therapeutic potential of targeting SIRT6 in pancreatic cancer. SIRT7, with its role in inhibiting EMT and suppressing metastatic potential, represents another valuable therapeutic target. Low nuclear levels of SIRT7 correlate with aggressive tumor phenotypes, making it a significant biomarker for disease prognosis. The therapeutic modulation of SIRT7 through inhibitors that stabilize its expression and block pro-tumorigenic pathways may provide a novel approach to controlling pancreatic cancer progression, particularly in cases resistant to conventional therapies.

In conclusion, targeting the sirtuin family, SIRT1—7, represents a comprehensive strategy for tackling pancreatic cancer’s complex biology. By addressing the distinct roles of each sirtuin in regulating apoptosis, metabolism, chemoresistance and metastasis, future research and therapeutic development could yield more effective and personalized treatments for this aggressive disease. The continued exploration of sirtuin modulation offers a promising direction for improving patient outcomes and overcoming the therapeutic challenges posed by pancreatic cancer. Table 2 encapsulates the innovative therapeutic strategies targeting the sirtuin family, each offering distinct potential to disrupt the key oncogenic processes in pancreatic cancer. The development of sirtuin-targeted drugs continues to be a promising frontier in improving clinical outcomes for this formidable disease.

## 4. Conclusions

The diverse roles of sirtuins in pancreatic cancer underline their pivotal importance as regulators of cancer metabolism, tumor progression, and chemoresistance. Each sirtuin contributes uniquely to the tumor microenvironment, influencing key oncogenic pathways and cellular processes. SIRT1, for instance, not only modulates apoptotic pathways and enhances cell survival but also plays a crucial role in chemoresistance, particularly through its interaction with hypoxic exosomal circRNAs like circZNF91. This interplay sustains glycolytic metabolism, which drives resistance to chemotherapeutic agents like gemcitabine. Targeting SIRT1 to modulate these pathways holds significant therapeutic promise in overcoming the resistance mechanisms inherent to pancreatic cancer. Similarly, SIRT2 contributes to tumorigenesis by regulating KRAS acetylation and MYC stability, while influencing the inflammatory responses that shape the pro-tumor microenvironment. Inhibiting SIRT2 with selective compounds, such as NPD11033, could reduce tumor cell proliferation and disrupt oncogenic signaling. SIRT3, a tumor suppressor, regulates mitochondrial metabolism and oxidative stress, counteracting tumor progression. Its suppression by miRNA-708-5p promotes glycolysis, but restoring SIRT3 activity could reverse these metabolic shifts and re-sensitize tumors to chemotherapy. SIRT4 plays a crucial role in metabolic regulation by inhibiting aerobic glycolysis and promoting autophagy, thus suppressing pancreatic cancer growth. Therapeutic interventions like Entinostat, which upregulate SIRT4, offer promising strategies to curb pancreatic cancer progression. SIRT5, through its modulation of glutamine and glutathione metabolism, provides another avenue for targeted therapy. Activators like MC3138 may disrupt PDAC metabolic dependencies, enhancing its sensitivity to gemcitabine. SIRT6 is involved in inflammation, angiogenesis, and chemoresistance, making it a promising target for therapeutic inhibition. Inhibitors such as the allosteric compound **11e** selectively enhance histone acetylation, suppressing metastatic activity, while further modulators like JYQ-42 highlight its role in reducing tumor invasiveness. SIRT7, regulating EMT and metastasis, modulates aggressive phenotypes by repressing genes such as COL4A1 and SLUG. Another critical aspect of pancreatic tumorigenesis lies in the highly heterogeneous nature of pancreatic cancer, which necessitates a comprehensive investigation of how hypoxia and the tumor microenvironment collectively drive tumor development and treatment resistance. Central to this dynamic is the hypoxia–Sirtuin axis, which profoundly shapes tumor progression and metabolic adaptation. Under hypoxic conditions, SIRT1 stabilizes HIF-1α through deacetylation, amplifying glycolytic metabolism and chemoresistance. This effect is exacerbated by hypoxic exosomal circRNAs, such as circZNF91, which competitively bind to miR-23b-3p, elevating SIRT1 expression and creating a feedback loop that sustains glycolysis and gemcitabine resistance. In contrast, tumor-suppressive Sirtuins like SIRT3 and SIRT4 counteract these processes by reprogramming hypoxia-driven oncogenic pathways. SIRT3 restoration promotes mitochondrial function and apoptosis, disrupting the metabolic shifts induced by hypoxia. Meanwhile, SIRT4, as a downstream target of the epigenetic regulator UHRF1, antagonizes the HIF-1α-mediated transcription of glycolytic genes, thereby inhibiting glycolysis, tumor proliferation, and metastasis. These insights into sirtuins reveal their multifaceted roles in pancreatic cancer, making them critical targets for therapeutic strategies aimed at disrupting cancer metabolism, tumor progression, and chemoresistance.

Sirtuin-based therapeutic approaches, which mainly target cellular metabolism and epigenetic regulation, hold significant promise but demand further advancements via both preclinical and clinical investigations to address their pleiotropic effects and enhance specificity. Unlike immunotherapy, which activates the immune system but has limited efficacy in pancreatic cancer, or nanomedicine, which improves drug delivery but faces challenges in biodistribution, sirtuin modulators uniquely target tumor metabolism and signaling pathways. Moreover, to fully realize their therapeutic potential, it is crucial to also understand and address the possible limitations of sirtuin-based therapeutic approaches, which must be overcome for successful cancer treatment. Their pleiotropic effects, involving diverse and sometimes opposing roles in cellular pathways, pose a risk of unintended consequences. Additionally, achieving tissue-specific targeting remains a challenge, as sirtuins are ubiquitously expressed and regulate both tumor-promoting and tumor-suppressive processes. Moreover, the potential for adaptive resistance to sirtuin modulators highlights the need for combination strategies and robust biomarker-guided therapies.

Future research should focus on optimizing sirtuin-targeted therapies by refining their specificity and enhancing their efficacy in combination with conventional treatments. Investigating synergistic strategies, such as combining sirtuin modulators with chemotherapeutics or metabolic inhibitors, could significantly improve clinical outcomes, particularly in combating drug resistance. Furthermore, the role of sirtuins in regulating non-coding RNAs, hypoxic signaling and metabolic pathways demands deeper exploration, as these mechanisms are critical in sustaining the aggressive nature of pancreatic cancer. Personalized therapies tailored to individual tumor profiles, especially considering the distinct roles of each sirtuin, will be essential in advancing pancreatic cancer treatment. Additionally, there is an urgent need to advance preclinical and clinical investigations of sirtuin-based inhibitors to address their pleiotropic effects, optimize specificity, and evaluate their safety and efficacy in cancer treatment. Such efforts are crucial for translating promising preclinical findings into effective therapeutic strategies for resistant malignancies like pancreatic cancer. Considering this, we are optimistic that the ongoing development of sirtuin-based inhibitors and their potential clinical applications may help overcome these challenges and offer a promising avenue for improving therapeutic efficacy in this notoriously resistant malignancy.

## Figures and Tables

**Figure 1 cancers-16-04095-f001:**
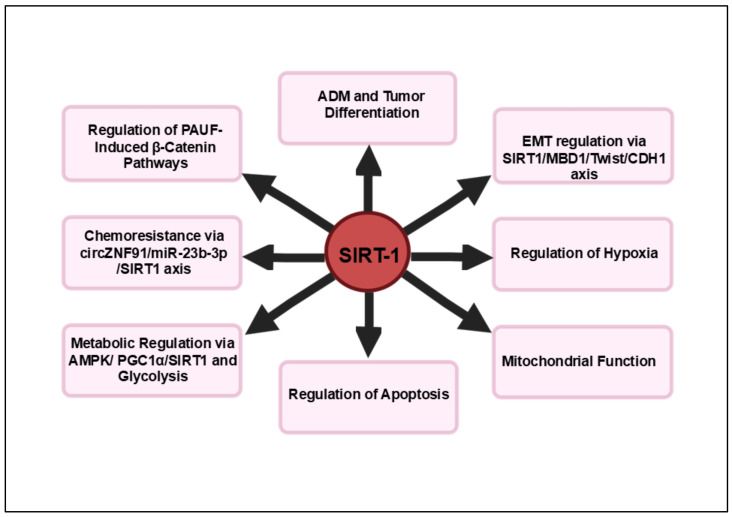
SIRT1 and pancreatic cancer. These pathways illustrate SIRT1’s broad impact on various processes critical for pancreatic cancer progression, including EMT, metabolic adaptation, survival under hypoxic conditions and resistance to apoptosis and chemotherapy. Targeting SIRT1 in these pathways holds therapeutic potential for overcoming drug resistance and improving treatment outcomes in pancreatic cancer.

**Figure 2 cancers-16-04095-f002:**
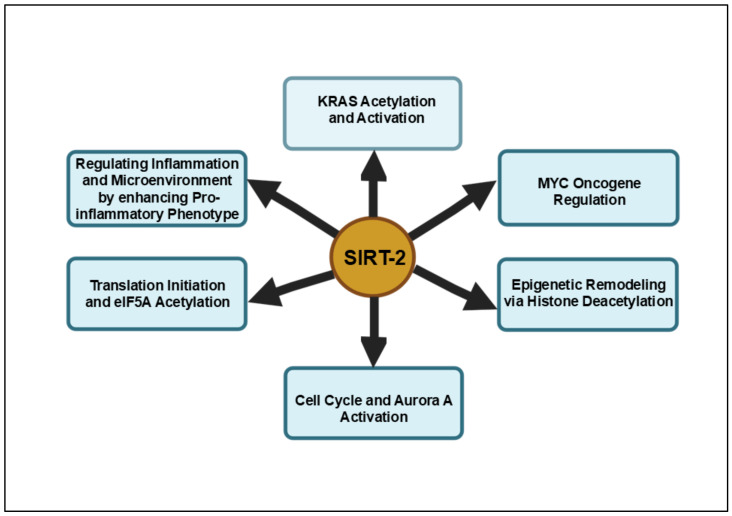
Summary of pancreatic cancer pathways influenced by SIRT2. These pathways highlight SIRT2’s multifaceted role in regulating KRAS-driven oncogenic signaling, MYC stability, inflammation and epigenetic remodeling, making it a promising therapeutic target in pancreatic cancer.

**Figure 3 cancers-16-04095-f003:**
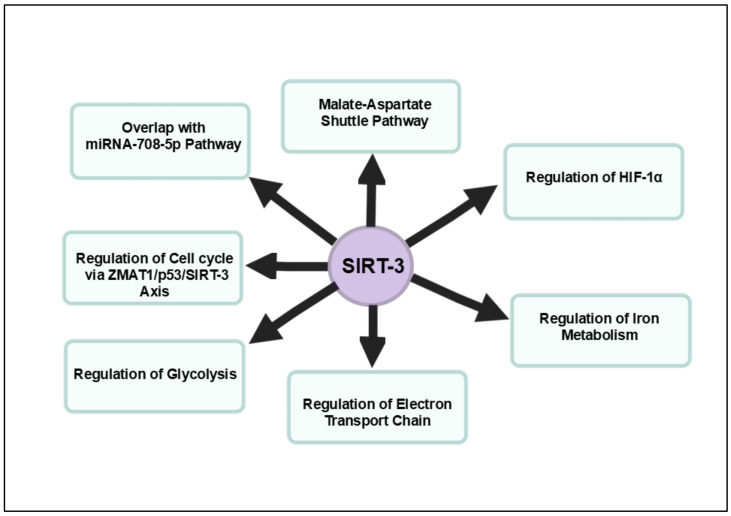
Overview of signaling pathways impacted by SIRT3 in pancreatic cancer. SIRT3 acts as a tumor suppressor by regulating mitochondrial function, oxidative stress and metabolic reprogramming in pancreatic cancer.

**Figure 4 cancers-16-04095-f004:**
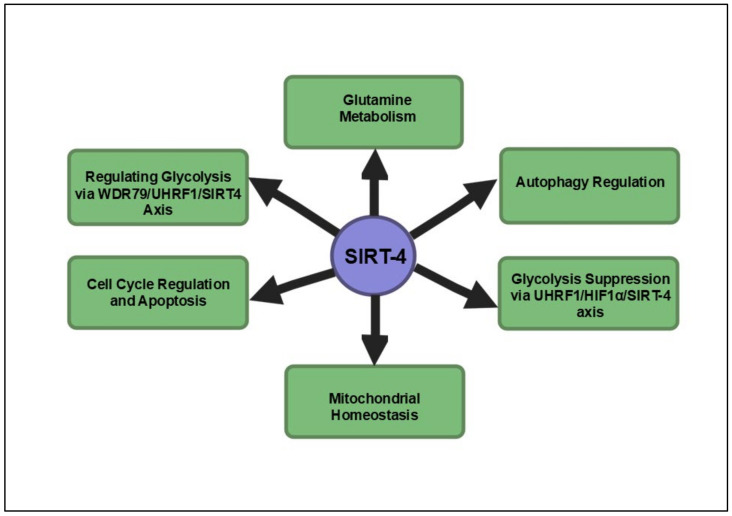
Summary of key signaling pathways influenced by SIRT4 in pancreatic cancer. These pathways collectively position SIRT4 as a critical tumor suppressor in pancreatic cancer, influencing metabolic reprogramming, autophagy and mitochondrial function.

**Figure 5 cancers-16-04095-f005:**
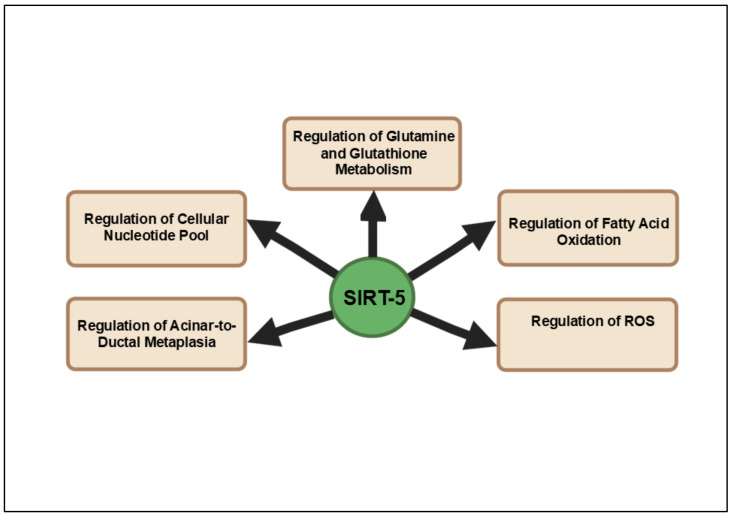
Overview of the principal signaling pathways affected by SIRT5 in pancreatic cancer. SIRT5 functions as a crucial metabolic gatekeeper in pancreatic cancer, with its modulation presenting potential therapeutic strategies to disrupt the metabolic dependencies of tumor cells.

**Figure 6 cancers-16-04095-f006:**
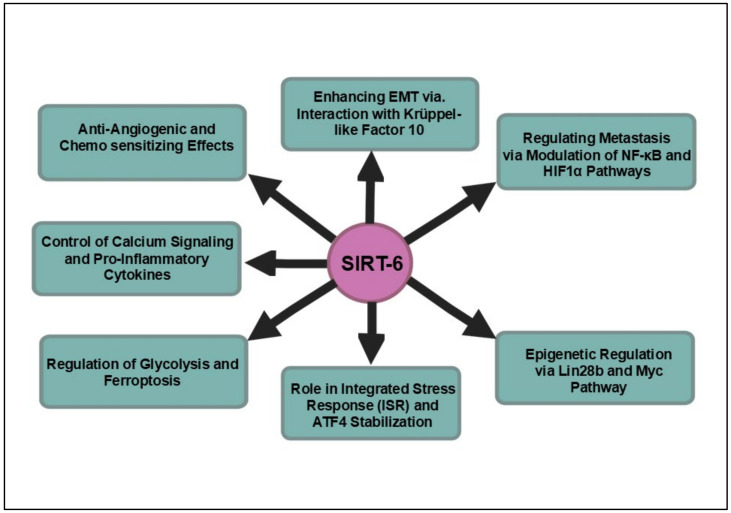
Overview of the major signaling pathways influenced by SIRT6 in pancreatic cancer. SIRT6 regulates critical pathways in pancreatic cancer, including the suppression of glycolysis, the modulation of the NF-κB and HIF1α axes and the stabilization of ATF4 for stress response. It also controls epigenetic regulation via Lin28b, impacting tumor growth and metastasis. Novel SIRT6 inhibitors show promise in enhancing chemotherapy sensitivity and inhibiting angiogenesis.

**Figure 7 cancers-16-04095-f007:**
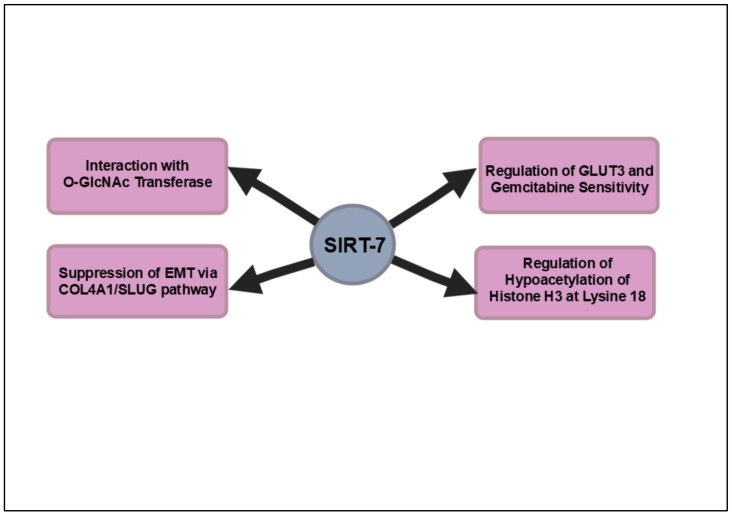
Overview of the key signaling pathways impacted by SIRT7 in pancreatic cancer. SIRT7 regulates pancreatic cancer progression by controlling GLUT3 expression, impacting gemcitabine sensitivity, and inhibiting EMT. O-GlcNAcylation stabilizes SIRT7, allowing it to repress tumor suppressor genes and promote tumor growth. Its low nuclear expression correlates with aggressive tumors, making SIRT7 a key prognostic biomarker and therapeutic target.

**Table 1 cancers-16-04095-t001:** Specific role of sirtuins in pancreatic cancer tumorigenesis and their potential as biomarkers for pancreatic cancer.

Sirtuin	Role in Pancreatic Tumorigenesis	Potential as Pancreatic Cancer Prognostic Biomarker	Reference
SIRT1	Expression was significantly higher in poorly differentiated carcinomas.	Yes	[134,135,136]
SIRT2	Expression is higher in carcinomas corelated with tumor progression.	Not yet established	[86,87,88]
SIRT3	Low expression contributes to tumor progression, development and poor prognosis.	Yes	[92,100,137,138]
SIRT4	Less expression in carcinomas contributes to escape from apoptosis and enhanced survival.	Not yet established	[109,111]
SIRT5	Loss of expression promotes tumorigenesis	Not yet established	[116]
SIRT6	Loss of expression is identified in basal PDAC, predicting worse survival.	Yes	[126,139,140]
SIRT7	Expression correlates with aggressive tumor phenotypes and poor patient outcomes	Yes	[79,93,132]

**Table 2 cancers-16-04095-t002:** Overview of existing sirtuin-targeting drugs in pancreatic cancer.

Sirtuin	Drug/Inhibitor	Mechanism of Action	Therapeutic Impact	Reference
SIRT1	Compound **19e**	Increases SIRT1 activity, promoting the deacetylation of apoptotic regulators.	Protects pancreatic β-cells from cytokine-induced apoptosis, reduces chemoresistance and enhances survival.	[84]
SIRT2	NPD11033	Selective small-molecule inhibitor, induces conformational change in SIRT2, blocking deacetylation.	Impairs PDAC cell proliferation, increases the acetylation of eIF5A and inhibits tumor growth.	[90]
SIRT3	Tris DBA (palladium complex)	Modulates autophagy and reduces mitochondrial ROS generation.	Re-establishes SIRT3 function, disrupts metabolic dysregulation and inhibits tumor progression.	[102]
SIRT4	Entinostat	Upregulates SIRT4 expression, inhibiting aerobic glycolysis.	Reduces PDAC growth and impairs cancer metabolism both in vitro and in vivo.	[112]
SIRT5	MC3138	Selective activator, mimics SIRT5 overexpression, enhancing acetylation.	Reduces nucleotide pools, sensitizes PDAC cells to gemcitabine and disrupts metabolic pathways.	[116]
SIRT6	Compound **11e**, pyrrole-pyridinimidazole derivatives, JYQ-42	Allosteric inhibitor, increases histone acetylation, inhibits angiogenesis.	Sensitizes PDAC cells to chemotherapy, reduces metastasis and suppresses tumor vascularization.	[129,130,131]
SIRT7	gemcitabine	Stabilizes SIRT7 expression, represses EMT-related gene expression.	Inhibits pancreatic cancer cell proliferation, migration and invasion, while reducing metastatic potential.	[127]

## Data Availability

Not applicable.

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
