# Peer review of "Sirtuins as Key Regulators in Pancreatic Cancer: Insights into Signaling Mechanisms and Therapeutic Implications"

_cancers, 2024, doi:10.3390/cancers16234095_

Round 1

Reviewer 1 Report

Comments and Suggestions for Authors

The manuscript titled "Sirtuins as Key Regulators in Pancreatic Cancer: Insights into Signaling Mechanisms and Therapeutic Implications" written by Surbhi Chouhan  and Anil Kumar presents a narrative review article that offers an  in-depth examination of the roles of sirtuins in pancreatic cancer, emphasizing their diverse functions in tumor metabolism, progression, and chemoresistance. While it highlights critical insights, some aspects could be ameliorated. Specifically : 

1) Increase the discussion related to preclinical or clinical trial outcomes to support claims about the therapeutic potential of sirtuins and address the complexities and risks associated with targeting sirtuins, such as their pleiotropic effects and potential for unintended consequences.

2) Focus on the most critical aspects of sirtuin biology and prioritize therapeutic strategies with the highest translational potential and include a discussion of how targeting sirtuins compares with other emerging therapeutic approaches for pancreatic cancer, such as immunotherapy or nanomedicine.

Moreover the figure quality is poor and not catchy and does not add any valuable clarification of the text. On the contrary the unique table added is useful and other tables may be added to summarize other aspects related to sirtuin activity. For example which of them is used as a biomarker of specific pathological condition or other aspects.

Author Response

Author’s Point by Point Response

Reviewer 1: The manuscript titled "Sirtuins as Key Regulators in Pancreatic Cancer: Insights into Signaling Mechanisms and Therapeutic Implications" written by Surbhi Chouhan and Anil Kumar presents a narrative review article that offers an in-depth examination of the roles of sirtuins in pancreatic cancer, emphasizing their diverse functions in tumor metabolism, progression, and chemoresistance.

Author’s reply- Thank you for your thoughtful and constructive feedback on our manuscript. We appreciate your insights, which will help refine our work and enhance its scientific value.

While it highlights critical insights, some aspects could be ameliorated. Specifically:

1) Increase the discussion related to preclinical or clinical trial outcomes to support claims about the therapeutic potential of sirtuins and address the complexities and risks associated with targeting sirtuins, such as their pleiotropic effects and potential for unintended consequences.

Author’s reply to comment 1- We appreciate the reviewers’ suggestion to incorporate preclinical or clinical trial outcomes to strengthen the claims regarding the therapeutic potential of sirtuins. There are few reports that emphasize clinical importance of sirtuins as potential biomarkers in pancreatic cancers which we have now summarized in a new table (Highlighted section: Heading: 3.SIRT-targeted drug responses in pancreatic cancer, Table. 1, lines-778). However, we would like to highlight that while sirtuins and their inhibitors have primarily been explored as modulators of signaling pathways in pancreatic cancer, it is noteworthy that none of these inhibitors have yet progressed to clinical trials. Therefore, we have now emphasized the rational and urgent need for advancing both preclinical and clinical investigations in the revised manuscript (Highlighted section: Heading: 4.Conclusion and perspectives, lines-909-916).

Furthermore, we recognize the importance of reviewer suggestions regarding risk associated with sirtuin based approaches and we think that it can be a great point to discuss. We have now elaborated this in Conclusion and perspectives section in revised manuscript to specifically address possible the complexities associated with targeting sirtuins, such as their pleiotropic effects and potential for adverse outcomes, ensuring a balanced perspective on their therapeutic applications. (Highlighted section: Heading: 4.Conclusion and perspectives, lines-886-893).

2) Focus on the most critical aspects of sirtuin biology and prioritize therapeutic strategies with the highest translational potential and include a discussion of how targeting sirtuins compares with other emerging therapeutic approaches for pancreatic cancer, such as immunotherapy or nanomedicine.

Author’s reply to comment 2- To streamline the discussion, we have now focused on the most significant aspects of sirtuin biology, particularly their roles in metabolic reprogramming, tumor progression and chemoresistance (Highlighted section: Heading: 4.Conclusion and perspectives, lines-841-885).  Additionally, we have now incorporated a discussion of potential therapeutic strategies with high translational potential and provided a dicussion of sirtuin-targeting approaches alongside emerging therapies, such as immunotherapy (Highlighted section: Heading: 4.Conclusion and perspectives, lines-886-892)

However, therapeutic approaches involving nanomedicine and it evaluation have not yet been explored in the context of sirtuin-based strategies. Nonetheless, we agree with the reviewer that recognizing the potential of nanomedicine to significantly impact the future of therapeutic regimens in pancreatic cancer is important. Additionally, we have now also added discussion regarding potential limitations of sirtuin-based approaches in pancreatic cancer treatment. We have now included this discussion in the Conclusion and perspectives section (Highlighted section: Heading: 4.Conclusion and perspectives, lines-892-899). This addition contextualizes the role of sirtuins within the broader landscape of pancreatic cancer treatment. We hope that the revisions made throughout indicated sections in the manuscript adequately address the reviewer’s concerns.

Moreover, the figure quality is poor and not catchy and does not add any valuable clarification of the text. On the contrary the unique table added is useful and other tables may be added to summarize other aspects related to sirtuin activity. For example, which of them is used as a biomarker of specific pathological condition or other aspects.

Author’s reply to comment - We now addressed the reviewer concern regarding the figure quality, and we have replaced the figures with high-resolution versions in the updated manuscript. Our objective in graphical illustration is to ensure the figures provide meaningful clarification and a concise summary of the roles of each individual sirtuin in pancreatic cancer. We have now made diligent efforts to make all figures more visually engaging while offering valuable insights. (lines: 253, 332, 465, 537, 571, 685,750)

Additionally, we have now included a new Table (Table 1 in the revised manuscript) that summarizes the specific roles of sirtuins in pancreatic tumorigenesis and identifies potential sirtuins as biomarkers for pancreatic cancer. (Highlighted section: Heading: 3.SIRT-targeted drug responses in pancreatic cancer, Table. 1, lines-778). We believe that these enhancements are intended to improve the clarity and utility of the manuscript for readers. We hope that these updates adequately address the reviewer’s concerns.

Reviewer 2 Report

Comments and Suggestions for Authors

Sirtuins are a family of NAD+-dependent deacetylases that regulate critical cellular processes involved in pancreatic cancer progression. SIRT1 promotes glycolytic metabolism under hypoxia and HIF-1α signaling, which affects apoptosis and chemotherapy resistance. SIRT2 inhibits eIF5A to interfere with cancer cell proliferation. SIRT3 modulates mitochondrial ROS and glycolysis to exert tumor-suppressive effects. SIRT4 suppresses aerobic glycolysis, and its therapeutic upregulation can control PDAC progression. SIRT5 regulates glutamine and glutathione metabolism, which may disrupt PDAC metabolic dependencies. SIRT6 and SIRT7 participate in angiogenesis, epithelial-mesenchymal transition (EMT), and metastasis. Therefore, precise modulation therapies targeting sirtuin activity should be developed to improve therapeutic efficacy and optimize the prognosis of patients with pancreatic malignancies.

The manuscript demonstrates a commendable level of writing and can be accepted with minor revisions. The necessary revisions are outlined below.

1. The author can provide a concise overview of the common targets of pancreatic cancer, as well as conventional chemotherapy drugs and targeted therapies, in the introduction section.

2. The authors should emphasize the impact of sirtuins as a potential therapeutic target for pancreatic cancer on patient prognosis, as well as the clinical significance of this target for drug development and research on therapeutic strategies.

3. The highly heterogeneous nature of pancreatic cancer necessitates a comprehensive examination of the impact that hypoxia processes and tumor microenvironment have on both tumor development and treatment. In light of this, the authors can explore the intricate relationship between these targets and the aforementioned factors.

4. The following articles are recommended:

[1] A. Gu, J. Li, S. Qiu, S. Hao, Z.-Y. Yue, S. Zhai, M.-Y. Li, Y. Liu, Pancreatic cancer environment: From patient-derived models to single-cell omics. Mol. Omics 2024, 20, 220233. DOI: 10.1039/D3MO00250K

[2] Olga Khersonsky, Sarel J. Fleishman. What Have We Learned from Design of Function in Large Proteins?. BioDesign Res. 2022;2022:9787581.DOI:10.34133/2022/9787581

[3] M. Chen, Y. Sun, H. Liu, Interdiscip. Med. 2023, 1, e20220012. https://doi.org/10.1002/INMD.20220012

Author Response

Author’s Point by Point Response

Reviewer comment: Sirtuins are a family of NAD+-dependent deacetylases that regulate critical cellular processes involved in pancreatic cancer progression. SIRT1 promotes glycolytic metabolism under hypoxia and HIF-1α signaling, which affects apoptosis and chemotherapy resistance. SIRT2 inhibits eIF5A to interfere with cancer cell proliferation. SIRT3 modulates mitochondrial ROS and glycolysis to exert tumor-suppressive effects. SIRT4 suppresses aerobic glycolysis, and its therapeutic upregulation can control PDAC progression. SIRT5 regulates glutamine and glutathione metabolism, which may disrupt PDAC metabolic dependencies. SIRT6 and SIRT7 participate in angiogenesis, epithelial-mesenchymal transition (EMT), and metastasis. Therefore, precise modulation therapies targeting sirtuin activity should be developed to improve therapeutic efficacy and optimize the prognosis of patients with pancreatic malignancies.

The manuscript demonstrates a commendable level of writing and can be accepted with minor revisions. The necessary revisions are outlined below.

Author’s reply- Thank you for your feedback. We're glad to hear that you found the review well-written. Your insights are valuable for refinement of our manuscript.

  1. The author can provide a concise overview of the common targets of pancreatic cancer, as well as conventional chemotherapy drugs and targeted therapies, in the introduction section.

Author’s reply to comment 1- We agree with reviewer’s suggestion, and we have now revised the introduction section to providing a concise overview of the common therapeutic targets, conventional chemotherapy drugs, and targeted therapies in pancreatic cancer. (Highlighted section: Heading: 1. Introduction, lines: 57-67, lines: 82-102).

  1. The authors should emphasize the impact of sirtuins as a potential therapeutic target for pancreatic cancer on patient prognosis, as well as the clinical significance of this target for drug development and research on therapeutic strategies.

Author’s reply to comment 2- We agree with reviewers’ suggestion, and we have now made rigorous efforts to provide a detailed analysis of the impact of sirtuins in pancreatic cancer, emphasizing their clinical significance as a biomarker for prognosis of pancreatic cancer for drug development and therapeutic strategies. (Highlighted section: Heading: 3.SIRT-targeted drug responses in pancreatic cancer, lines: 756-777, Table 1. line: 778).

However, we would like to highlight here that till date, sirtuins and their inhibitors have primarily only been investigated as modulators of signaling cascades in pancreatic cancer and none of these inhibitors advancing clinical trials for pancreatic cancer. We certainly acknowledge reviewer point and recognize that that can be a great topic to discuss as potential for future research in this area. We have now expanded on this topic in the Future Directions sections along with discussing possible limitations of sirtuin-based therapeutic approaches in pancreatic cancer (Highlighted section: Heading: 4.Conclusion and perspectives, lines: 909-916).

  1. The highly heterogeneous nature of pancreatic cancer necessitates a comprehensive examination of the impact that hypoxia processes and tumor microenvironment have on both tumor development and treatment. In light of this, the authors can explore the intricate relationship between these targets and the aforementioned factors.

Author’s reply to comment 3- We agree with the reviewer’s opinions on the highly heterogeneous nature of pancreatic cancer. As the literature on individual sirtuins in the context of tumor heterogeneity, hypoxia mediated by HIF1-α, and the broader tumor microenvironment, continues to grow, we have incorporated these aspects under the sections discussing the roles of individual sirtuins in pancreatic cancer in the revised manuscript (Highlighted section: lines: 228-238, 262-269, 286-291, 369-383, 397-413, 416-425, 502-511, 603-610, 645-657, 732-735).

Also, In the revised manuscript, we have now elaborated on this in the Conclusion and perspectives section, emphasizing how these factors influence tumor progression and therapy resistance (Highlighted section: Heading: 4.Conclusion and perspectives, lines: 868-884). This addition aims to provide a more comprehensive perspective on the therapeutic challenges and opportunities in pancreatic cancer.

  1. The following articles are recommended:

[1] A. Gu, J. Li, S. Qiu, S. Hao, Z.-Y. Yue, S. Zhai, M.-Y. Li, Y. Liu, Pancreatic cancer environment: From patient-derived models to single-cell omics. Mol. Omics 2024, 20, 220–233. DOI: 10.1039/D3MO00250K

[2] Olga Khersonsky, Sarel J. Fleishman. What Have We Learned from Design of Function in Large Proteins?. BioDesign Res. 2022;2022:9787581.DOI:10.34133/2022/9787581

[3] M. Chen, Y. Sun, H. Liu, Interdiscip. Med. 2023, 1, e20220012. https://doi.org/10.1002/INMD.20220012

Author’s reply to comment 4- We have now incorporated relevant insights and appropriately cite relevant reference in the revised manuscript. (Highlighted section: Heading: 5. Reference, line: 1027, 1038, 1041). We appreciate your efforts to provide constructive feedback and are confident that these revisions will strengthen the manuscript's impact and clarity.